# Root-Knot Nematode Early Infection Suppresses Immune Response and Elicits the Antioxidant System in Tomato

**DOI:** 10.3390/ijms252312602

**Published:** 2024-11-23

**Authors:** Sergio Molinari, Anna Carla Farano, Paola Leonetti

**Affiliations:** 1Bari Unit, Institute for Sustainable Plant Protection, Department of Biology, Agricultural and Food Sciences, National Research Council of Italy, 70126 Bari, Italy; paola.leonetti@cnr.it; 2Sant’Anna School of Advanced Studies, 56127 Pisa, Italy; annacarla.farano@santannapisa.it

**Keywords:** antioxidant system, JA/ET-signaling, plant immune system, *PR*-genes, root-knot nematodes, ROS, SA-signaling, tomato

## Abstract

The immune response in plants is regulated by several phytohormones and involves the overexpression of defense genes, including the pathogenesis-related (*PR-*) genes. The data reported in this paper indicate that nematodes can suppress the immune response by inhibiting the expression of defense genes. Transcripts from nine defense genes were detected by qRT-PCR in the roots of tomato plants at three and seven days post-inoculation (dpi) with living juveniles (J2s) of *Meloidogyne incognita* (root-knot nematodes, RKNs). All the salicylic acid (SA)-responsive genes tested (*PR-1*, *PR-2*, *PR-4b*, *PR-5*) were down-regulated in response to nematode infection. On the contrary, the expression of jasmonic acid (JA)-responsive genes, including *ACO* (encoding the enzyme 1-aminocyclopropane-1-carboxylic acid oxidase, which catalyzes the last step of ethylene (ET) biosynthesis) and *JERF3* (*Jasmonate Ethylene Response Factor 3*), was unaffected by the infection. Conversely, the effect of nematode attack on the activities of the defense enzymes endoglucanase and endochitinase, encoded by *PR-2* and *PR-3,* respectively, changed depending on the tested dpi. At 5 dpi, both enzymes were inhibited in inoculated plants compared to healthy controls. The genes encoding glutathione peroxidase (*GPX*) and catalase (*CAT*), both part of the antioxidant plant system, were highly overexpressed. Additionally, the activity of the antioxidant enzymes superoxide dismutase (SOD), CAT, and ascorbate peroxidase (APX) was enhanced in infected roots. Isoelectrofocusing of root extracts revealed novel SOD isoforms in samples from inoculated plants. Furthermore, plants were pre-treated with an array of key compounds, including hormone generators, inhibitors of SA or JA-mediated defense pathways, reactive oxygen species (ROS) scavengers and generators, inhibitors of ROS generation, and compounds that interfere with calcium-mediated metabolism. After treatments, plants were inoculated with RKNs, and nematodes were allowed to complete their life cycle. Factors of plant growth and infection level in treated plants were compared with those from untreated inoculated plants. Generally, compounds that decreased SA and/or ROS levels increased infection severity, while those that reduced JA/ET levels did not affect infection rates. ROS generators induced resistance against the pests. Compounds that silence calcium signaling by preventing its intake augmented infection symptoms. The data shown in this paper indicate that SA-mediated plant immune responses are consistently suppressed during the early stages of nematode infection, and this restriction is associated with the activation of the antioxidant ROS-scavenging system.

## 1. Introduction

Plants adopt sophisticated molecular mechanisms to activate immune responses against pathogens and parasites. The plant immune response is regulated by several phytohormones, including salicylic acid (SA), jasmonic acid (JA), and ethylene (ET). However, adapted pathogens can evade plant defenses by delivering effector molecules directly into plant cells. Endoparasitic sedentary nematodes (ESNs), such as root-knot nematodes (RKNs) belonging to *Meloidogyne* spp., produce proteins in their esophageal glands, which they inject into root cells via a stylet. A growing body of evidence shows that most of these proteins are effectors that can suppress plant defense during infection [1,2,3,4,5]. Different effectors use different mechanisms to impair plant defense responses, which, in turn, may differ according to plant species and the nematode family [6]. The success of the nematode in suppressing plant defenses depends on the stage of infection, which involves a complex, intimate interaction with the root tissue. RKNs enter the roots as motile second-stage juveniles (J2s) and move intercellularly to reach the vascular cylinder. They introduce their stylet into specific parenchyma cells and induce the formation of feeding sites by transforming those cells into giant or nurse cells. These feeding sites supply the developing nematode with solutes and nutrients. J2s become sedentary, molting through the J3 and J4 stages to develop into adult females. At the end of their lifecycle, they lay eggs in an external gelatinous matrix that becomes clearly visible outside the roots as an egg mass. Moreover, nematode activity induces hypertrophy and hyperplasia in the surrounding tissues, causing the formation of characteristic galls on the roots [7]. We investigated whether nematodes could suppress plant defenses from the earliest stages of infection, specifically during the initial formation of feeding sites. We focused on the interaction between *M. incognita* and tomato (*Solanum lycopersicum* L.) because it is one of the most studied interactions in molecular nematology. Tomato is one of the most widely cultivated vegetables worldwide because of its high nutritive value and multiple uses, and *M. incognita* is its most damaging nematode pest with high impacts on tomato yields. However, most studies on ESN-induced suppression of host immune response have been carried out on *Arabidopsis thaliana*, largely due to the availability of mutants defective in key immunity-regulating genes [3,8]. Nematode effectors in *Arabidopsis* giant cells and galls result in an extensive down-regulation of gene expression, particularly of stress-related genes [9,10].

In this paper, we examine the expression of stress-related genes, regulated by both SA- and JA/ET-mediated defense pathways, in whole tomato roots at the earliest stages of nematode infection. Activation of hormonal signaling pathways, such as SA, JA, and ET pathways, may trigger either local or systemic immune responses. The SA pathway is primarily induced against biotrophic pathogens, such as ESNs and sap-sucking aphids, while the JA pathway is primarily induced against herbivores and necrotrophic pathogens, such as chewing/mining insects and endoparasitic migratory nematodes [11]. A crosstalk exists between SA- and JA-signaling, resulting in reciprocal antagonism [12]. SA is known to suppress the JA pathway, although ET signaling prevents SA-mediated suppression of the JA pathway [13]. SA-JA crosstalk may be interpreted as a plant strategy aiming at fighting different pests/pathogens with different defense responses in a cost-saving mode. Generally, induced defense activates a wide range of reactions expressed in morphological changes and up-regulation of defense genes, such as those encoding for chitinase, protease inhibitors, etc., or production of toxins, such as phytoalexins, alkaloids, and glucosinolates, or release of volatiles that attract predators or parasitoids in the case of insect herbivore attacks [13]. 

Many studies already investigated nematode manipulation of plant hormone pathways [14] as well as gene and RNA silencing [15,16], both well-known mechanisms in RKN parasitism. Less well-known is that nematodes suppress plant immune responses by activating the host reactive oxygen species (ROS)-scavenging system and/or inhibiting ROS generation [1,17]. When plant immunity against RKNs is fully activated, a marked and prolonged production of ROS, such as superoxide anion (O_2_^−●^) and hydrogen peroxide (H_2_O_2_), occurs. ROS are directly toxic to the invading juveniles, although they have been proven to be signaling molecules for additional immune outputs that contrast nematode survival and development in roots [18]. Suppression of ROS is, then, mandatory for a successful infection by nematodes. It has already been reported that one of the earliest events in RNK-tomato compatible interaction is the inhibition of the NADPH oxidases that are primarily involved in ROS production as they transfer electrons from cytosolic NADPH to apoplastic oxygen thus generating O_2_^−●^ [17]. By the enzyme superoxide dismutase (SOD), superoxides are rapidly converted into H_2_O_2_, which is as toxic but more stable than O_2_^−●^ and capable of crossing the lipid bilayer of the plasma membrane [19]. The activation of H_2_O_2_-degrading enzymes, such as ascorbate peroxidase (APX), glutathione peroxidase (GPX) and catalase (CAT), is presumably one of the targets for the early actions of invading nematodes. In particular, CAT, a tetrameric heme-containing enzyme found in all aerobic organisms, is a key enzyme in converting high concentrations of intracellular H_2_O_2_ in H_2_O and O_2_. The activities of such enzymes and the number of transcripts of the encoding genes have herein been detected in roots during the early stages of nematode infection and confronted with those from unaffected roots.

Finally, we describe changes in the expression of key genes and enzyme activities related to the root antioxidant system. Moreover, we treated plants with several compounds that interfere with this antioxidant system before inoculating them with nematodes. At the end of the nematode life cycle, we compared plant growth and infection severity of treated plants with those of untreated controls. Our findings indicate that nematodes use multi-target strategies to contrast plant basal defense as they attempt to build feeding sites in roots; the two main early targets identified were the inhibition of SA-dependent defense pathways and the maintenance of low ROS levels, alongside the activation of the antioxidant system.

## 2. Results 

### 2.1. Role of Defense-Mediating Plant Hormones in Nematode–Tomato Interaction 

Solutions of SA, JA, and ethylene generators, such as MetSA, MetJA, and ethephon, respectively, were sprayed onto the green parts of tomato plants 1 day before inoculation with RKNs. Growth and infection factors of treated (sprayed) plants were detected 7 weeks after inoculation and confronted with those of plants inoculated with RKNs but not sprayed (Table 1). Only treatment with the SA-generator MetSA markedly reduced both nematode reproduction rates (RP) and root galling of untreated plants, measured as sedentary forms per gram of root fresh tissue (SFs g^−1^rfw); the relief of symptoms resulted in enhanced shoot growth. Neither MetJA nor ethephon reduced the infection level of untreated control plants.

Furthermore, we tested the role of two important inhibitors of hormone biosynthesis in nematode–plant interaction. One was salicyl-hydroxamic acid (SHAM), an inhibitor of lipoxygenases (LOXs), which are key enzymes in JA biosynthesis [20]; the second was paclobutrazol (PCB), an inhibitor of SA synthesis used to reduce SA levels [21]. When SHAM was applied to plants as a pre-treatment before nematode inoculation, it inhibited nematode reproduction and root galling, indicating that JA is needed for a fully successful compatible RKN–plant interaction. Conversely, when the SA level was lowered by PCB treatment, the already weak basal defense was further hampered, allowing the nematodes to even enhance their reproduction rate (Table 2).

### 2.2. Suppression of Plant Immune Response by Root-Knot Nematodes

The involvement of SA- and JA/ET-dependent defense pathways in the early stages of nematode infection was also tested by monitoring the expression of various genes involved in these pathways. The expression of the SA-dependent pathogenesis-related (*PR-*) genes *PR-1*, *PR-4b*, *PR-2,* and *PR5*, which encode PR-proteins that act as the executioners of plant immunity, was detected in roots of tomato plants at 3 and 7 dpi and compared with that of non-inoculated control plants (Figure 1). At 3 dpi, all the tested genes were down-regulated in roots (Figure 1A), suggesting that the SA-dependent defense pathway is repressed by J2s in the earliest stages of infection. Conversely, at 7 dpi, when most feeding sites were already formed, repression was relieved (Figure 1B).

Expression of a set of genes, considered part of the JA/ET-dependent defense pathway, was detected at 3 and 7 dpi in the roots of plants uninoculated and inoculated with RKNs (Figure 2). The set included *PR-3*, which encodes an endochitinase; *Jasmonate Ethylene Response Factor 3* (*JERF3*), which is a JA/ET-responsive gene; and the *1-aminocyclopropane-1-carboxylic acid oxidase* gene (*ACO*), which encodes for the ACO enzyme involved in the last step of ET biosynthesis. Expression of *JERF-3* and *ACO* genes was not affected in the early steps of nematode infection at 3 dpi. In contrast, *PR-3* was overexpressed at 3 dpi but down-regulated at 7 dpi (Figure 2A,B).

Consequently, we tested the activity of the enzymes encoded by the SA-responsive gene *PR-2*, i.e., β-1,3-glucanase, and by the JA/ET-responsive gene *PR-3*, i.e., endochitinases. Tested enzymes were extracted from the cytoplasmic fraction of roots from control and inoculated plants at 3, 5, and 7 dpi (Table 3).

The activity of both glucanase and endochitinases was inhibited in infected roots only at 5 dpi. At 3 and 7 dpi, glucanase activity was unaffected, whileendochitinase activity was enhanced. 

### 2.3. Elicitation of the Plant Antioxidant System by Root-Knot Nematodes

Glutathione peroxidase (GPX) and catalase (CAT) are antioxidant enzymes capable of actively scavenging ROS and mitigating inflammatory responses in plant tissue. Plants generate ROS to defend themselves against biotic challenges, although Figure 3 shows that *GPX* and *CAT* gene expressions were highly augmented during the early stages of compatible RKN-tomato interaction. 

Therefore, we tested the activity of CAT and two other key antioxidant enzymes: SOD, an active scavenger of superoxide radicals (O_2_^−●^) to form hydrogen peroxide (H_2_O_2_), and APX, an active scavenger of H_2_O_2_, like CAT (Table 4).

All these root enzyme activities increased from 3 to 7 dpi compared to uninfected plants, with the exception of SOD at 7 dpi. 

Electrophoresis separation of CAT and SOD isozymes, based on charge, revealed two additional neutral SOD isozymes in root protein extracts from infected plants at 5 dpi compared to those from uninfected plants. No changes were observed in the number of CAT isozyme bands between infected and uninfected extracts; however, basic CAT bands appeared much more marked in infected samples than in uninfected ones (Figure 4).

Since it was evident that developing nematodes promotes the antioxidant enzyme system for the establishment of functional feeding sites, we provided plants with inhibitors of key enzymes of ROS metabolism and then infected them with RKNs. Growth and infection factors were monitored at the end of the first generation’s life cycle. The inhibitors tested were: (i) diphenyliodonium chloride (DPI), a suicide inhibitor of the O_2_^●−^-generating NADPH oxidase [22]; (ii) N-N’-dimethylthiourea (DMTU), a ROS scavenger [23]; (iii) 3-amino-1,2,4-triazole (3-AT), a known inhibitor of CAT; (iv) sodium diethyl-dithiocarbamate-trihydrate (DIECA), a strong inhibitor of SOD that causes O_2_^●−^ accumulation [24] (Table 5). DIECA and 3-AT, which raise ROS levels, induced a resistant response to nematodes, markedly decreasing their reproduction rate and root galling compared with uninfected plants. DPI and DMTU, which lower ROS levels, increased infection symptoms in treated plants.

### 2.4. Role of Ca^2+^-Dependent Metabolism in Nematode–Tomato Interaction

Key metabolic inhibitors were used to study the role of Ca^2+^-dependent metabolism in nematode–tomato interaction. One was okadaic acid (OKA), a type 1 and 2A protein phosphatases inhibitor and a known tumor promoter; OKA has been shown to block SA-mediated induction of the *PR-1* gene and decrease Ca^2+^ influx into the cytoplasm [25,26]. The second compound, D-sphingosine (SFI), is reported to inhibit calmodulin-dependent enzymes and Ca^2+^ influx in cells [27]. The third compound was a Ca-ionophore that disrupts Ca^2+^ gradients across cell membranes. Plants were treated by overnight root dip in aqueous solutions of these compounds, while control plants were dipped in the same solutions without compounds. The following day, all plants were inoculated with RKNs. After 7 weeks, growth and infection factors were measured (Table 6).

The compounds that hamper Ca^2+^ signal increased nematode infection symptoms except Ca-ionophore. Plant growth parameters were not significantly affected by treatments.

## 3. Discussion

It is now generally recognized that RKNs are able to suppress the plant immune system through the injection of an array of effectors directly into the cells and/or by secretion from cuticlin or amphids in the root apoplasm [1,2,3,4]. A down-regulation of the SA-responsive *PR* genes in galls and giant cells of infested roots is one of the effects of this suppression [9,10]. Our data indicate that this down-regulation involves the whole roots, at least at the earliest stages of nematode infection. It is likely that RKNs may induce the inhibition of SA biosynthesis to down-regulate SA-responsive *PR*-gene expression. Chorismate mutase, which is one of the suppressors secreted by RKNs, can divert the chorismate pool from SA biosynthesis to pre-phenate synthesis, which leads to decreased SA levels in cells and apoplasm. Chorismate mutase has been found to be secreted also by two migratory endoparasitic nematodes, *Hirsmanniella oryzae* and *Pratylenchuscoffeae*, thus suggesting that inhibition of SA synthesis may be a process conserved to suppress host defenses by phytoparasitic nematodes [4]. SA-mediated defenses are specifically and promptly raised as soon as the invading juveniles start to build feeding sites. In our study, the chemical inhibition of SA synthesis by PCB prior to inoculation made plants more sensitive to nematode attack, thus proving that the decrease in SA level is a target of the initial phases of nematode invasion. On the contrary, when plants were exogenously provided with MetSA, infection was almost halved in terms of reproduction rates and root galling. 

It is known that JA-signaling in *Arabidopsis* acts in wounding response synergistically with ABA via transcription factor *AtMYC2*, whereas it acts together with ET via transcription factor *ERF1* (*Ethylene Response Factor*) during pathogen attacks. The expression of *ERF1* plays a role in the expression of defense genes, such as *PR-4*, *PR-3,* and *PDF1.2* [28]. We detected the expression of the *ACO* gene, which encodes for the enzyme involved in the last step of ET biosynthesis, and *JERF3*, a novel member of tomato *ERF* transcription factors that responds to both JA and ET signaling by binding to ET/JA responsive GCC box [29]. Both *JERF3* and *ACO* gene expressions seem not to change in the early stages of nematode infection with respect to healthy plants; this may indicate that both JA and ET are not involved in the infection steps in which SA is involved. Comparably, if plants are pre-treated before nematode inoculation with MetJA and ethephon, which generate JA and ET, respectively, no apparent significant change is observed in infection standard factors. On the contrary, when plants were pre-treated with SHAM, an inhibitor of JA biosynthesis, nematode infection was reduced, thus indicating that JA is needed for a fully successful compatible RKN–plant interaction. RKNs, after entering the roots, have an initial migratory phase of parasitism to reach the vascular cylinder, during which wounding signals are released and JA signaling is activated. Motile juveniles may have to contrast JA-mediated defense reactions by secreting specific suppressors. However, once they establish their feeding structure, or in order to build it, they need to reverse their action and activate a functional JA signaling pathway [30]. Accordingly, at the end of the experimental period (7dpi), the activity of the JA-responsive defense enzyme endochitinase in inoculated roots was found to be higher than that in control roots. Conversely, the SA-responsive defense enzyme glucanase had its activity unaffected at the same infection stage.

The role of ET in plant-nematode interactions is controversial; most studies report an inhibition of RKN infection by ET caused by a decrease in nematode attraction to the roots [14]. Moreover, induced resistance by pre-treatments with BCAs was associated with *ACO* gene up-regulation with a predicted increase in ET level [31]. However, resistance to RKNs is considered to be related to an increased sensitivity to ET rather than to an increased ET level [32]. The calreticulin effector (Mi-CRT), a CRT calcium-binding protein synthesized in the subventral glands of *M. incognita* preparasitic J2 and in the dorsal gland of parasitic stages and secreted into root cells, was proved to impair plant basal defense by suppressing the induction of *PDF1.2*, an ET-responsive defense gene [3]. Our data indicate that if ET is provided by ethephon pre-treatment to plants, there is no repression of nematode infection; rather, EM numbers were found to be higher in pre-treated than in control plants. Also, expression of the *ACO* gene was found not to change after infection, thus suggesting that ET-mediated signaling may not be involved in compatible plant-nematode interaction, at least at the studied stages. Therefore, suppression of plant immune response by RKNs should mainly be addressed to inhibiting the SA-dependent defense pathway, in agreement with the paradigm that SA protects against biotrophic pests, such as all endoparasitic sedentary nematodes (ESNs), whereas JA protects from necrotrophic microorganisms and pests, such as endoparasitic migratory nematodes and munching insects [33].

Another nematode effector, named MjTTL5, was reported to suppress plant immune response by interacting with an *Arabidopsis* key component of the antioxidant system (AtFTRc); such an interaction could drastically increase the host ROS scavenging activity, resulting in suppression of plant basal defense and support to nematode infection [1]. An additional target of the early nematode action is then the enhancement of the antioxidant ROS scavenging system in order to lower the inflammatory defense reaction of plants to leave undisturbed their tumor-like activity on the roots. In the early stages of nematode infection, either the expression of antioxidant genes, such as GPX and CAT, or the activity of antioxidant enzymes, such as SOD, CAT, and APX, have herein been found to increase. Isoelectrofocusing of protein extracts confirmed that bands of basic CAT isoforms were more stained in samples coming from roots of infected plants; likewise, two additional neutral SOD isoforms were found in the same samples with respect to samples from uninfected roots. As the ROS-scavenging system is activated, the ROS-generating system is inhibited. The gene encoding for cytochrome P450, which is a key factor in microsomal O_2_^●−^production, was reported to be down-regulated in the earliest stages of infection to soybean by soybean cyst nematodes [34]. Comparably, the O_2_^●−^generating NADPH oxidase activity was found to be inhibited in the microsomes of tomato plants infected with RKNs [17]. Moreover, if we strengthen the inhibition of this enzyme by providing the suicide inhibitor DPI to plants before inoculation—in other words, if we help nematodes to further inhibit ROS production by the roots—the result is an augmentation of infection level in treated inoculated plants. Damages to plants were even worse whenDMTU, another ROS scavenger, was provided before nematode inoculation. For the successful development of nematodes in roots, H_2_O_2_ levels and peroxidative reactions must be maintained at the lowest rates during the early stages of nematode infection to impede cell oxidative stress and cell death, which normally occur in incompatible interactions. Probably, also alternative oxidase, which promotes mitochondrial CN-resistant NADH oxidation and functions as a mechanism to decrease ROS formation, is induced upon nematode attack on susceptible plants [35,36]. On the contrary, any metabolite or metabolic pathway that increases ROS level and promotes an inflammatory reaction contributes to lowering the severity of infection to the point of obtaining a marked relief of the symptoms. We planned to increase ROS levels in roots by pre-treating plants with the inhibitors SHAM, 3-AT, and DIECA; SHAM raises ROS levels by inhibiting CN-resistant respiration, while 3-AT and DIECA inhibit the antioxidant enzymes CAT and SOD, respectively. In plants pre-treated with those inhibitors, symptoms of infection were consistently slighter than those of untreated plants. On the other hand, inhibition of CAT has been proposed to have a key role in both genetic (*Mi-1.2*-carrying) and induced resistance of tomato against *Meloidogyne* spp. [17,31,37].

The oxidation burst by which plants generally react to biotic stresses, nematode attack included, depends on the activation of the NADPH oxidase named RBOHD, which belongs to the respiratory burst oxidase homolog (RBOH) family and is deputed to cell death control, cell-wall damage-induced lignification, and systemic defense signaling [38]. It has already been mentioned that nematode action is addressed to maintain this activity low and obtain a successful compatible interaction. Regulation of this enzyme depends on protein kinase activities and a continuous Ca^+2^-influx, which is one of the earliest events in plant immune response [38]. It is reasonable that nematodes need to impair this Ca^+2^-influx and operate on protein kinase activities, thus causing the reported inhibition of RBOHD. RBOHD in microsomes isolated from tomato roots has been reported to be inhibited by exogenous Ca^2+^ [19]. Calreticulins (CRTs),which are highly conserved calcium-binding proteins and important Ca^2+^ chelators, are injected in the apoplasm of roots by RKNs to prevent Ca^2+^ influx into the cells, which may initiate an immune response [39]. The deletion of the Ca^2+^ gradient existing between apoplasm and cytoplasm may be obtained by the production of natural Ca^2+^-ionophores by the action of nematodes on cell membranes. When the artificial Ca^2+^-ionophore A23187 was provided to plants before inoculation, no evident effect was produced on the infestation level; this might indicate that artificial Ca^2+^-ionophores cause a deletion of Ca^2+^ gradient similar to the one normally provoked by nematodes during the early phases of their attack. Comparably, when plants were pre-treated with compounds, such as SFI and OKA, which inhibit protein kinase activities and probably cause additional inhibition of RBOHD, both nematode reproduction rate and root galling significantly increased with respect to untreated plants. These findings reveal that, although we have used known susceptible tomato cvs, there is always a weak basal defense reaction against aggressive nematode populations that is normally overcome by invading juveniles. Figure 5 shows the possible mechanisms by which RKNs suppress the tomato immune system activated as a first basal defense reaction during the earliest phases of nematode infection. However, this basal defense reaction may be weakened or strengthened according to different pre-treatments with specific compounds or by enriching the soil with a beneficial rhizosphere microbiome [40]. 

The knowledge of the molecular strategies adopted by nematodes to circumvent the defense responses of plants is crucial for promoting more sustainable nematode management. The aim should be the arrangement of treatments with non-toxic chemicals and/or beneficial microorganisms that may activate the immune system so as to retard and/or reduce the suppressive action of nematodes and make plants more tolerant to their infection without using or with a substantial reduction in chemical nematicides, which are highly toxic to all soil biota and consumers. The prolonged use of toxic chemicals has led to serious problems, such as the resistance to conventional nematicides, reduction in the ecosystem biodiversity, environmental contamination, and low soil productivity, which compelled the scientific community to investigate more reliable and sustainable control strategies, such as those foreseen in microbiome-assisted agriculture [41]. The contrast to nematode attack may be helpful in reducing the severity of the concomitant aboveground pest diseases, as it has been shown that nematode infection lowers the potential of defense responses also in leaves. Therefore, activation of the plant immune system should be promoted in Integrated Pest Management (IPM) strategy that will make plants more responsive to most environmental challenges and open the way to more sustainable nematode control, as indicated by scientific European communities.

## 4. Materials and Methods

### 4.1. Chemical Treatments of Tomato Plants 

Experiments involved five tomato (*Solanum lycopersicum* L.) cultivars: Roma VF, Regina, Fiaschetto, Principe Borghese, and Marmande, all confirmed as susceptible to RKNs. Seeds were sown in special containers filled with sterilized topsoil at 23–25 °C in a glasshouse. Seedlings were transferred to 110 cm^3^ clay pots filled with freshly field-collected loamy soil and put on temperature-controlled benches (soil temperature maintained at 23–25 °C) located inside the glasshouse. Plantlets were provided with a regular regime of 12 h light/day, watered with Hoagland’s solution, and allowed to grow to the 4–6 compound leaf stage (reaching a biomass of approximately 3–4 g). One day before nematode inoculation, groups of plants were designated for treatment with different chemicals, and other groups were left untreated as controls. 

The set of hormone-like compounds was applied as water-based sprays to the green parts of plants, while controls were sprayed with distilled water. To prevent soil contamination, the surface of the soil in each pot was covered with aluminum foil. Hormone-like compounds were applied at optimal dosages:-Met-SA at 17.5 mg per plant-Met-JA at 12.5 mg per plant-ethephon at 3.1 mg per plant

The rest of the chemicals were administered to plants by immersing the roots in chemical water solutions overnight, while controls were immersed in distilled water. The day after, the roots were thoroughly washed, and plants were immediately inoculated with nematodes. Each batch of 6 plants was immersed in 200 mL of the designated chemical solution.

The sets of key enzyme inhibitors, metabolic pathway inhibitors, and calcium metabolism effectors were provided as follows:-DPI at 0.6 mg per plant-DMTU at 10 mg per plant-3-AT at 0.03 mg per plant-DIECA at 0.9 mg per plant-SHAM at 0.75 mg per plant-PCB at 3.0 mg per plant (PCB was dissolved in ethanol, with minimal amounts of mother solution added to 200 mL of water)-SFI and OKA at 0.04 mg per plant (both were dissolved in DMSO, with minimal amounts of mother solutions added to 200 mL of water)-Ca-ionophore A23187 at 0.3 mg per plant

All chemicals were purchased from Sigma-Aldrich (Milan, Italy)

### 4.2. Nematode Inoculation and Detection of Infection

Field populations of the RKN *Meloidogyne incognita* (Kofi *et* White) Chitw. were reared on susceptible tomato plants in a glasshouse and used for inoculation. Egg masses from heavily infested roots were hand-picked under a stereoscope, put on 500-mesh filters in glass Petri dishes half filled with tap water for J2 hatching, and incubated at 25 °C in the dark. J2 was collected for at least two days after incubation. J2 concentration per mL of suspension was determined under a stereoscope at ×25 magnification. Both chemical-treated and untreated groups of plants were inoculated with 400 active motile juveniles (J2s) per plant. Inoculation was carried out by pouring 2–4mls of J2s stirring suspensions into two holes made at the base of plants. Inoculations occurred one day after chemical treatment. This one-day interval before inoculation was allowed because chemicals might be absorbed by roots and make their effects on plants before the nematode attack. Inoculations of untreated susceptible tomato plants with reared nematode populations gave severe standard infections, which were used to establish a standard level of infection severity for comparison with that resulting from chemical treatments of plants.

The effects of chemical treatments on nematode infection and plant growth were detected 7 weeks after RKN inoculation, at the end of the nematode lifecycle. At this time from inoculation, the inoculated J2s, after two molts in J3-J4s, developed in gravid females and were able to reproduce. Each treatment was tested on at least six plants. At the end of the experimental time, plants were harvested, and roots were gently washed to remove soil residues and separated from shoots. The measured growth metrics were shoot (SW) and root fresh weights (RW). 

Under the adopted experimental conditions, a fraction of the 400 inoculated J2s completed their life cycle in one month, depositing eggs in gelatinous egg masses (EMs) outside the roots. The newly hatched J2s re-infested roots as a second generation, with a much larger population than the inoculated one. In our experiments, second-generation J2s could only enter the roots and develop into sedentary forms (SFs: J3s, J4s, swollen females), but adult females do not have enough time to produce eggs for reproduction. Therefore, reproduction metrics such as egg masses per gram of root fresh weight (EMs g^−1^rfw), female fecundity (FF), and reproduction potential (RP) apply only to the first generation. Root damage, primarily due to galling, is quantified by the numbers of sedentary forms per gram of root fresh weight (SFs g^−1^rfw), which are largely determined by the second generation. Reproduction Potentials (RPs) reflect the multiplication of the initial nematode population by the end of the experiment, whereas female fecundity (FF) is the average number of eggs contained in a single EM. 

To extract the different nematode life stages, roots were chopped into 2 cm segments. Two root systems were sampled to create three weighed subsamples. One subsample was stained by immersion in a 0.1 g/L solution of Eosin Yellow for at least 1 h in a refrigerator; red-colored EMs were then counted under a stereoscope (×6 magnification). The other twosubsamples were processed to extract eggs and SFs according to the techniques described in [42].Once extracted, SFs and eggs were counted under a stereoscope at ×12 and ×25 magnification, respectively. RP was calculated as RP = *P_f_/P_i_*, where *P_i_* (initial population) is the number of J2s inoculated in each pot, and *P_f_* (final population) is the total number of eggs per root system (J2s free in soil were negligible in the small pots used in the experiments). FF was calculated as total eggs/EMs.

### 4.3. Protein Extraction and Enzyme Activity Assays

Proteins were extracted from the roots of plants harvested 3, 5, and 7 days after nematode inoculation. Roots were separated from shoots and thoroughly rinsed with tap water to remove soil residues. Then, they were dried, weighed, and either put on ice for protein extraction or stored at −80 °C. Extraction started with grinding of the roots in porcelain mortars by immersion in liquid nitrogen. Three different samples of roots from non-inoculated and three from inoculated plants were ground for each bioassay. Ground powder was suspended in a buffer (1:5 *w*/*v*, pH 6.0) consisting of 0.1 M potassium phosphate, 4% polyvinylpyrrolidone, and the protease inhibitor phenylmethanesulfonyl fluoride (PMSF, 1 mM). Suspensions were ground further using a Polytron^®^ PT–10–35 (Kinematica GmbH, Malters, Switzerland) and filtered through four layers of gauze. Filtrates were centrifuged at 12,000× *g* for 15 min. Supernatants were filtered through 0.45 µm nitrocellulose filters applied to 10 mL syringes. Finally, they were ultra-filtered at 4 °C through 2-mL Vivaspin micro-concentrators (10,000 molecular weight cutoff, Sartorius Stedim, Biotech GmbH, Jena, Germany). Retained protein suspensions were used in protein content and enzyme assays. Protein content was determined through the enhanced alkaline copper protein assay with bovine serum albumin as the standard [43].

Superoxide dismutase (SOD) activity was assessed using a spectrophotometric method based on the inhibition of cytochrome *c* reduction. The assay was conducted in a total reaction volume of 1 mL, which included root extract (10–20 µL) as the source of SOD, 20 µM cytochrome *c*, and a system for generating superoxide radicals composed of 1 mM xanthine and 20 mU xanthine oxidase. To stabilize the reaction, 20 mM sodium azide (NaN_3_) and 0.5 mM ethylenediaminetetraacetic acid (EDTA) were added to the reaction mixture, which was prepared in 0.1 M Na-K-phosphate buffer, pH 7.8. Xanthine oxidase addition started the reaction, and cytochrome reduction was monitored at 550/540 nm in a Perkin Elmer 557 double-beam spectrophotometer. The SOD activity was calculated as the percentage of inhibition of cytochrome c reduction, where one unit of enzyme produced 50% inhibition in the standard reaction setup [44]. 

Catalase activity (CAT) was assessed by measuring the initial rate of hydrogen peroxide (H_2_O_2_) disappearance [45]. The reaction was conducted in a total volume of 0.5 mL, consisting of 20 mM H_2_O_2_ and 10–20 µL of root extracts in 0.1 M sodium phosphate buffer, pH 7.0. H_2_O_2_ oxidation was monitored by recording the decrease in the absorbance at 240 nm. One unit of enzyme reduced 1 mmole of H_2_O_2_ per minute (ε = 0.038 mM^−1^ cm^−1^).

Ascorbate peroxidase activity (APX) was determined by measuring the rate of ascorbate oxidation in the presence of H_2_O_2_ [46]. The reaction mixture was prepared in a final volume of 0.5 mL, containing 0.1 M TES buffer (pH 7.0), 0.1 mM EDTA, 1 mM ascorbate, 0.1 mM H_2_O_2_, and 10–20 µL root extracts. Ascorbate oxidation was recorded as the decrease in absorbance at 298 nm and one unit of enzyme oxidized 1 µmole ascorbate min^−1^ (ε = 0.8 mM^−1^ cm^−1^).

Chitinase activity (CHI) was measured using a colorimetric procedure that detects N-acetyl-D-glucosamine (NAG) [47]. Chetobiose produced by the hydrolytic action of chitinase is converted into NAG by the β-glucuronidase introduced in the reaction mixture. The reaction mixture was prepared in a final volume of 0.5 mL, containing suspended chitin (10 mg/mL) from shrimp shells (Sigma-Aldrich, Italy), 100 µL of root extract, and 0.05 M sodium acetate buffer (pH 5.2) containing 0.5 M NaCl. Suspensions were incubated in an orbital incubator at 37 °C for 1 h, after which the reaction was stopped by boiling at 100 °C for 5 min in a water bath. Following centrifugation at 10,000× *g* for 5 min at room temperature, supernatants (300 µL) were collected and added with 5 µL of β-glucuronidase (Sigma, type HP-2S, 9.8 units mL^−1^). The reaction was again stopped by boiling, followed by cooling at room temperature. After adding 60 µL of 0.8 M potassium tetraborate (pH 9.1), mixtures were heated to 100 °C for 3 min and cooled to room temperature. Finally, 1.2 mL of 1% 4-dimethylaminobenzaldehyde (DMAB, Sigma) was added, and the mixtures were incubated at 37 °C for 20 min. Absorbance was read at 585 nm (DU-70, Beckman, Brea, CA, USA), and the amount of NAG produced was determined using a standard curve generated with known concentrations (4.5–90 nmoles) of commercial NAG (Sigma-Aldrich, Italy). In negative controls, tissue extracts were not added, while positive controls included 10 µL of chitinase from *Streptomyces griseus* (Sigma, 200 units/g). Chitinase activity was expressed as nanokatal mg^−1^ protein (nkat mg^−1^prot) and one nkat defined as 1.0 nmol of NAG produced per second at 37 °C.

β-1,3-endoglucanase (glucanase, GLU) activity was determined by measuring the amount of glucose released from the substrate laminarin (Sigma, Italy). The reaction mixtures (0.4 mL) consisted of laminarin (0.4 mg), 100 µL of tissue extracts, and 0.1 M sodium acetate buffer (pH 5.2). Mixtures were incubated at 37 °C for 30 min for glucose production. Afterward, 300 µL of Nelson alkaline copper reagent was added, and the mixtures were heated to 100 °C for 10 min, followed by cooling at room temperature. Once cooled, 100 µL of Nelson chromogenic reagent was added for reducing sugar assays [48]. Negative controls contained grinding buffer instead of root extracts, and positive controls included laminarinase (2 U/mL). Enzyme activity was expressed as µmol of glucose equivalents released per minute according to a standard curve created with known amounts (10–200 µg mL^−1^) of commercial glucose (Sigma, Italy).

### 4.4. Electrophoresis Procedure andEnzyme Bands Staining

Isozymes of CAT and SOD were separated by isoelectric focusing on mini-gels (3.6 cm separation zone) with a Phast-System^®^ (GE Healthcare Life Science, Milan, Italy), as already described [49]. Root extracts were further concentrated using Amicon Ultra-0.5 mL centrifugal filters (Merck Life Science, Milan, Italy) at 9000× *g* for 30 min to achieve approximately 70 µg of protein (8 µL) per sample, which was then loaded into individual wells of isoelectric focusing (IEF) gels. Gels were calibrated using a broad pI calibration kit (GE Healthcare Life Science, Italy) covering a pI range of 3.5 to 9.3. 

IEF gels were stained for CAT by soaking in 0.6% H_2_O_2_ for 60 s and rinsing with distilled water. Afterward, gels were placed in Petri dishes containing a staining solution of 0.2% (*w*/*v*) potassium iodide and 0.1% (*v*/*v*) glacial acetic acid. Under these conditions, CAT bands appeared transparent against a dark yellow background. Gels were immediately drained and photographed before background fading occurred. Digital photos were later converted to grayscale for analysis.

SOD bands were stained by incubating the gels in the dark at 37 °C for 5 min in a solution containing 0.12 mM nitrobluetetrazolium (NBT). Then, gels were rinsed with distilled water and placed in Petri dishes containing a solution of 15 mM N,N,N’,N’-tetramethyl-ethylenediamine (TEMED) and 0.26 mM riboflavin in 0.1 M K-phosphate buffer (pH 7.8). Petri dishes were positioned on a white light trans-illuminator until white bands of SOD activity appeared against a blue background; as this coloration is stable over time, gels were dried and directly scanned to obtain digital images. 

### 4.5. RNA Extraction, cDNA Synthesis, and Quantitative Real-Time Polymerase Chain Reaction 

Roots from non-inoculated and nematode-inoculated plants were collected and weighed at 3 and 7 dpi. Root samples were ground in liquid nitrogen, and 100 mg macerated tissue was used for each extraction of total RNA using the RNA-easy Plant Mini Kit (Qiagen, Hilden, Germany) according to the manufacturer’s protocol. RNA integrity was tested by electrophoresis runs on 1.0% agarose gel and quantified using a nano-drop spectrophotometer. Total RNA (1 μg) was used for cDNA synthesis with a QuantiTect Reverse Transcription Kit (Qiagen, Germany) with random hexamers. qRT-PCR reactions were prepared to a final volume of 20 μL using RNAse-free water, 0.2 μM each of forward and reverse primers, 1.5 μL cDNA template, and 10 μL SYBR^®^ Select Master Mix (Applied Biosystems, Monza, Italy). Amplification involved an initial denaturation step at 95 °C (10 min), followed by 40 cycles at 95 °C (30 s), at 58 °C (30 s), at 72 °C (30 s), with a final extension step at 60 °C (1 min). qRT-PCRs were performed in triplicate using the Applied Biosystems^®^StepOne™ (ThermoFisher Scientific, Monza, Italy). The accession numbers, the corresponding proteins, and the primer sequences for the tested genes are reported in Table 7. *Actin-7* (NM_001308447.1, *ACT-7*) served as the reference gene for quantification, as its expression levels were stable after inoculation.Gene transcript levels were expressed as 1/ΔC_t_, where ΔC_t_ is the difference in cycle thresholds (C_t_) between the tested gene and the reference gene (*Actin-7*).

### 4.6. Experimental Design and Statistical Analysis

Three different bioassays were carried out to collect data on plant growth and infection factors in untreated control plants and in plants treated with each compound. In each bioassay, six untreated and six treated plants were used; three replicates for the tested factors were obtained from each bioassay. Reported factor values are the means obtained from 9 replicates (*n* = 9). Means ± standard deviations of values from control plants were separated from those of treated plants by a paired *t*-test (significance levels: * *p* < 0.05). 

Enzyme activity data were collected at 3, 5, and 7 dpi from both non-inoculated and inoculated plants. Two independent bioassays, each using 12 plants per treatment, were carried out. For each bioassay, three protein extractions were performed. Each protein extract was measured three times for each enzyme activity. Means (*n* = 18) ± standard deviations were compared using a paired *t*-test (significance levels: * *p* < 0.05). 

Gene expression data were collected at 3 and 7 dpi from non-inoculated and inoculated plants. Two independent bioassays, each with 12 plants per treatment, were carried out. Three separate RNA extractions were conducted for each treatment of each bioassay, followed by qRT-PCRs in triplicate. Means (*n* = 18) ± standard deviations of 1/ΔC_t_ (ΔC_t_ = C_t_ target gene—C_t_ actin) were analyzed using paired *t*-tests (significance levels: **p* < 0.05; ***p* < 0.01). Statistical analysis was conducted in MS Excel.

## Figures and Tables

**Figure 1 ijms-25-12602-f001:**
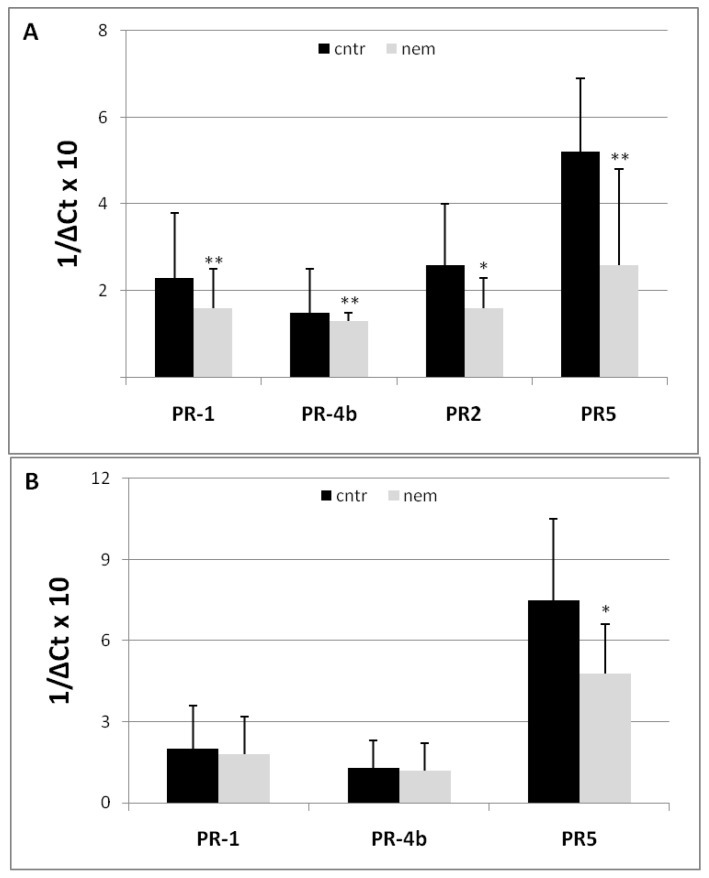
Expression of SA-dependent genes in tomato roots 3 (**A**) and 7 (**B**) days after RKN inoculation (nem) and in non-inoculated roots (cntr). Gene expression was detected by quantitative real-time reverse-transcription polymerase chain reaction (qRT-PCR). Gene transcript levels are expressed as 1/ΔC_t_, where ΔC_t_is the difference between the cycle threshold (C_t_) of the tested gene and that of the reference gene (*Actin 7*). Higher 1/ΔC_t_ values indicate higher gene expression. Values are expressed as means (*n =* 9) ± standard deviations. Means from inoculated roots were compared with their controls using a paired *t*-test (* *p* < 0.05; ** *p* < 0.01).

**Figure 2 ijms-25-12602-f002:**
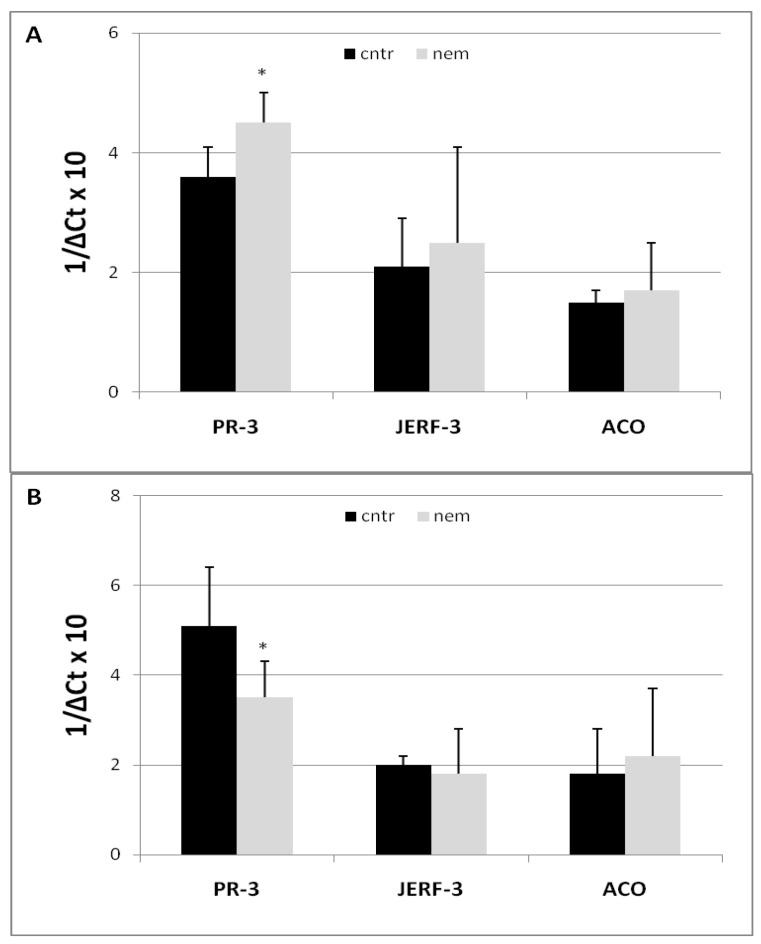
Expression of *PR-3*, *JERF-3*, and *ACO* JA-ET dependent genes in tomato roots 3 (**A**) and 7 (**B**) days after RKN inoculation (nem) and non-inoculated roots taken as controls (cntr). Expression of genes was detected by quantitative real-time reverse-transcription polymerase chain reaction (qRT-PCR). Gene transcript levels are expressed as 1/ΔC_t_, where ΔC_t_ is the difference between the cycle thresholds (C_t_) of the tested gene and that of the reference gene (*Actin 7*). Higher 1/ΔC_t_ values indicate higher gene expression. Values are expressed as means (*n =* 9) ± standard deviations. Means from inoculated roots were compared with their controls using a paired *t*-test (* *p* < 0.05).

**Figure 3 ijms-25-12602-f003:**
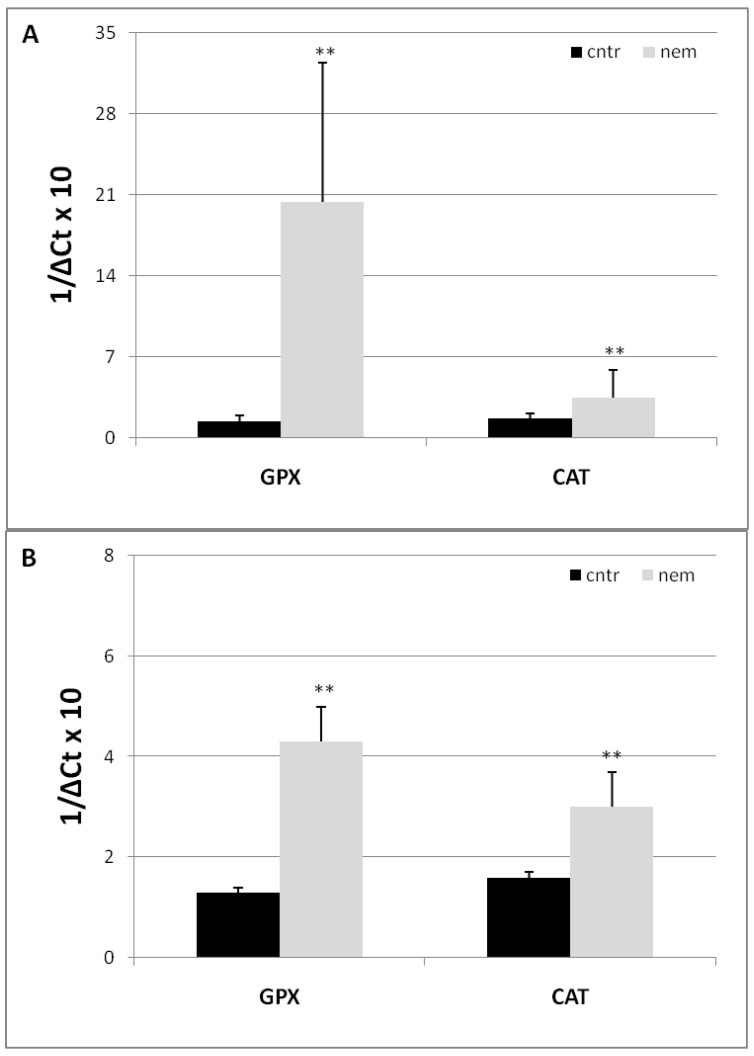
Expression of *GPX* and *CAT* ROS-scavenging genes in tomato roots 3 (**A**) and 7 (**B**) days after RKN inoculation (nem) and non-inoculated roots taken as controls (cntr). Expression of genes was detected by quantitative real-time reverse-transcription polymerase chain reaction (qRT-PCR). Gene transcript levels are expressed as 1/ΔC_t_, where ΔC_t_ is the difference between the cycle threshold (C_t_) of the tested gene and that of the reference gene (*Actin 7*). Higher 1/ΔC_t_ values indicate higher gene expression. Values are expressed as means (*n =* 9) ± standard deviations. Means from inoculated roots were compared with their controls using a paired *t*-test (** *p* < 0.01).

**Figure 4 ijms-25-12602-f004:**
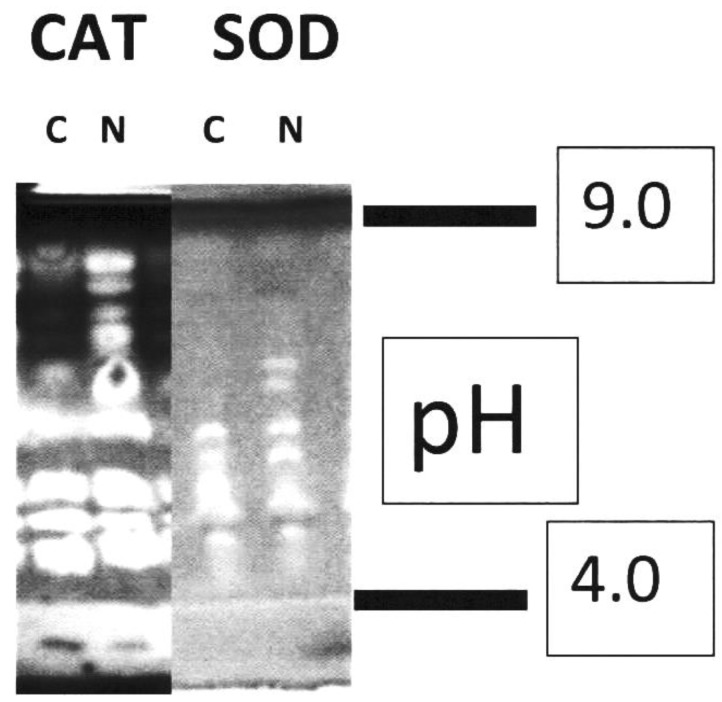
Isoelectrofocusing of root extracts stained for catalase (CAT) and superoxide dismutase (SOD). Root extracts were obtained from non-inoculated (C) and inoculated (N) plants at 5 days post-inoculation with RKNs. Digital images of mini-gels (3.6 cm separation zone) were adjusted to display white enzyme bands on a dark background. Gels were calibrated using a broad pI calibration kit containing proteins with pI values ranging from 3.5 to 9.3.

**Figure 5 ijms-25-12602-f005:**
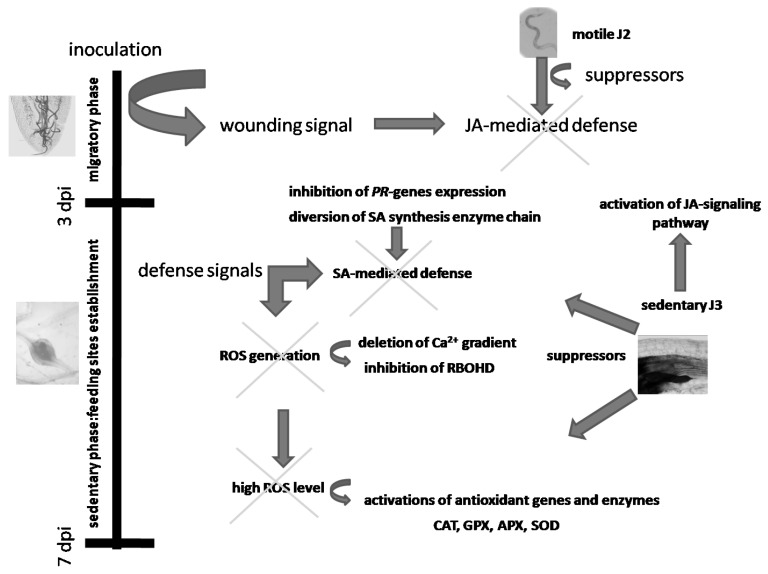
Scheme of the possible mechanisms by which RKNs suppress basal defense reactions of tomato at the earliest stages of infection. Arrows indicate the processes by which inhibitions/suppressions (X) of biochemical pathways occur.

**Table 1 ijms-25-12602-t001:** Effect of MetSA, MetJA, and ethephon spray on tomato plants inoculated with RKNs. Plant growth (shoot weight, SW, root weight, RW) and infection factors (EMs and SFs g^−1^, FF, RP)were measured in treated and untreated (Cntr) plants. Values are expressed as means (*n* = 9) ± standard deviations. Significant changes with respect to control values, determined by a *t*-test (*p* < 0.05), are indicated with an asterisk (*). Percentage differences in treated plants compared to untreated controls (Cntr) are indicated in (%).

	Cntr	MetSA	MetJA	Ethephon
SW	9.4 ± 3.7	11.8 ± 8.3 * (26)	8.6 ± 3.9	10.5 ± 2.9
RW	2.2 ± 1.5	2.3 ± 0.8	2.0 ± 1.3	2.5 ± 1.7
EMs g^−1^	32 ± 14	18 ± 13 * (−43)	37 ± 19	44 ± 21 * (39)
SFs g^−1^	174 ± 125	106 ± 67 * (−39)	171 ± 157	188 ± 100
FF	353 ± 128	353 ± 239	335 ± 131	289 ± 182
RP	108 ± 43	58 ± 52 * (−46)	92 ± 33	99 ± 49

**Table 2 ijms-25-12602-t002:** Effect of overnight root dipping in SHAM and PCB solutions on tomato plants inoculated with RKNs. Plant growth (SW, RW) and infection factors (EMs and SFs g^−1^, FF, RP) were measured in treated and untreated (Cntr) plants. Values are expressed as means (*n* = 9) ± standard deviations. Significant changes with respect to control values, determined by a *t*-test (*p* < 0.05), are indicated with an asterisk (*). Percentage differences in treated plants compared to untreated controls (Cntr) are indicated in (%).

	Cntr	SHAM	PCB
SW	12.3 ± 5.4	13.3 ± 6.0	12.7 ± 7.7
RW	2.5 ± 1.1	2.3 ± 1.5	2.4 ± 1.1
EMs g^−1^	43 ± 24	28 ± 15 * (−35)	55 ± 34 * (27)
SFs g^−1^	321 ± 187	180 ± 82 * (−44)	325 ± 168
FF	581 ± 189	598 ± 191	598 ± 55
RP	143 ± 71	103 ± 72 * (−28)	180 ± 60 * (26)

**Table 3 ijms-25-12602-t003:** Glucanase (GLU) and endochitinases (CHI) activities from cytoplasmic fractions in root extracts of control (Cntr) and RKN-inoculated (Nem) tomato plants. Enzyme activities were detected 3, 5, and 7 days post-inoculation (dpi). Significant changes, according to a *t*-test (*p* < 0.05), are indicated by an asterisk. Values are expressed as means (*n* = 6) ± standard deviations. Percentage differences in inoculated plants with respect to control (Cntr) are indicated in (%).

	3 dpi		5 dpi		7 dpi	
	Cntr	Nem	Cntr	Nem	Cntr	Nem
GLU ^a^	30.1 ± 6.1	28.1 ± 6.4	49.1 ± 11.6	38.0 ± 8.1 * (−37)	21.9 ± 7.8	25.5 ± 2.4
CHI ^b^	0.10 ± 0.02	0.15 ± 0.04 * (50)	0.26 ± 0.02	0.18 ± 0.02 * (−29)	0.11 ± 0.03	0.14 ± 0.02 * (26)

^a^ µmol glucose equivalents released min^−1^ mg^−1^prot; ^b^ nkat mg^−1^prot.

**Table 4 ijms-25-12602-t004:** Superoxide dismutase (SOD), catalase (CAT), and ascorbate peroxidase (APX) activities from cytoplasmic fractions in root extracts of control (Cntr) and RKN-inoculated (Nem) tomato plants. Enzyme activities were detected 3, 5, and 7 days post-inoculation (dpi). Significant changes, determined by a *t*-test (*p* < 0.05), are indicated with an asterisk (*). Values are expressed as means (*n* = 6) ± standard deviations. Percentage differences in inoculated plants with respect to control (Cntr) are indicated in (%).

	3 dpi		5 dpi		7 dpi	
	Cntr	Nem	Cntr	Nem	Cntr	Nem
SOD ^a^	34.6 ± 12.3	45.3 ± 22.0 * (31)	36.0 ± 25.0	48.0 ± 26.0 * (35)	12.3 ± 6.3	10.1 ± 5.4
CAT ^a^	8.5 ± 1.8	10.3 ± 3.9 * (22)	9.0 ± 2.1	13.9 ± 3.9 * (55)	10.0 ± 2.4	25.6 ± 9.1 * (157)
APX ^a^	0.15 ± 0.03	0.24 ± 0.03 * (55)	0.28 ± 0.06	0.36 ± 0.10 * (25)	0.17 ± 0.07	0.28 ± 0.04 * (65)

^a^ units mg^−1^ protein.

**Table 5 ijms-25-12602-t005:** Effect of overnight root dipping in DPI, DMTU, 3-AT, and DIECA solutions on tomato plants inoculated with RKNs. Plant growth (weight of shoots, SW; weight of shoots, RW) and infection factors (EMs and SFs g^−1^, FF, RP) were measured in treated and untreated (Cntr). Values are expressed as means (*n* = 9) ± standard deviations. Significant changes with respect to control values, determined by a *t*-test (*p* < 0.05), are indicated with an asterisk. Percentage differences in treated plants compared to untreated controls (Cntr) are indicated in (%).

	Cntr	DPI	DMTU	3-AT	DIECA
SW	9.0 ± 4.6	10.3 ± 3.9	12.2 ± 7.1 * (36)	7.7 ± 3.3	7.7 ± 6.3
RW	1.7 ± 0.9	1.9 ± 1.3	1.6 ± 1.1	1.2 ± 1.0 * (−25)	1.6 ± 1.3
EMs g^−1^	53 ± 33	64 ± 50 * (21)	70 ± 34 * (34)	34 ± 24 * (−36)	37 ± 27 * (−30)
SFs g^−1^	268 ± 152	308 ± 122	466 ± 261 * (74)	nd	163 ± 148 * (−39)
FF	428 ± 103	436 ± 187	nd	471 ± 142	475 ± 128
RP	140 ± 106	219 ± 204 * (56)	nd	55 ± 48 * (−61)	45 ± 51 * (−68)

nd = not determined.

**Table 6 ijms-25-12602-t006:** Effect of overnight root dipping in SFI, OKA, and Ca-ionophore solutions on tomato plants inoculated with RKNs. Plant growth (SW, RW) and infection factors (EMs and SFs g^−1^rfw) were measured in treated and untreated (Cntr) plants. Values are expressed as means (*n* = 9) ± standard deviations. Significant changes with respect to control values, determined by a *t*-test (*p* < 0.05), are indicated with an asterisk. Percentage differences in treated plants compared to untreated controls (Cntr) are indicated in (%).

	Cntr	SFI	OKA	Ca-Ionophore
SW	7.3 ± 2.3	7.6 ± 2.3	7.6 ± 2.0	8.3 ± 2.6
RW	1.3 ± 0.5	1.5 ± 1.0	1.3 ± 1.1	1.6 ± 0.6
EMs g^−1^	85 ± 37	116 ± 29 * (37)	99 ± 7	78 ± 32
SFs g^−1^	139 ± 33	185 ± 68 * (33)	204 ± 43 * (47)	157 ± 56

**Table 7 ijms-25-12602-t007:** Tomato defense-related genes examined in this study and the specific primers used in a quantitative reverse transcriptasepolymerase chain reaction (qRT-PCR).

Gene	Accession Number	Encoded Protein Activity	Primer Sequence
*PR1-1b*	NM_001247385.2	Unknown	F:GATCGGACAACGTCCTTAC R:GCAACATCAAAAGGGAAATAAT
*PR2*	NM_001247229	β-1,3-glucanase	F: AAGTATATAGCTGTTGTAATGAA R: ATTCTCATCAAACATGGCGAA
*PR-3*	NM_001247474.2	Chitinase	F:AACTATGGGCCATGTGGAAGA R:GGCTTTGGGGATTGAGGAG
*PR-4b/P2*	NM_001247154.1	Pathogenesis-related gene 4b	F: TGACCAACACAGGAACAGGA R: GCCCAATCCATTAGTGTCCA
*PR-5*	NM_001247422.3	Thaumatin-like	F:GCAACAACTGTCCATACACC R:AGACTCCACCACAATCACC
*JERF3*	NM_001247533.2	Jasmonate ethylene response factor 3	F:GCCATTTGCCTTCTCTGCTTC R:GCAGCAGCATCCTTGTCTGA
*ACO*	XM_015225653.2	1-aminocyclopropane-1-carboxylic acid oxidase	F:CCATCATTTCTCCAGCATCA R:TTGGCAGACTCAAATCTAGG
*CAT*	NM_001247257.2	Catalase 2	F:TGCTCCAAAGTGTGCTCATC R:TTGCATCCTCCTCTGAAACC
*GPX*	XM_004244468.3	Glutathione peroxidase	F: GTTTGCTTGCACACGGTTTA R: CGTCGTTGGTGGATACCTCT
*Actin 7*	NM_001308447.1	Actin	F:CAGCAGATGTGGATCTCAAA R:CTGTGGACAATGGAAGGAC

## Data Availability

The original contributions presented in this study are included in the article. Further inquiries can be directed to the corresponding author.

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
