# Peer review of "Root-Knot Nematode Early Infection Suppresses Immune Response and Elicits the Antioxidant System in Tomato"

_ijms, 2024, doi:10.3390/ijms252312602_

Round 1

Reviewer 1 Report

Comments and Suggestions for Authors

Review Report

This manuscript provides an in-depth examination of the immune responses in tomato plants to M. incognita (root-knot nematodes, RKNs), focusing on the expression of defense genes, antioxidant activity, and the impact of specific compounds on infection severity. The study highlights the role of salicylic acid (SA)-mediated immune responses and the suppression of defense gene expression upon nematode infection, contributing valuable insights to plant-pathogen interaction research.

Comments for authors

The introduction of this manuscript provides a foundational overview of the mechanisms through which root-knot nematodes (RKNs) interact with and suppress plant immune responses, focusing specifically on their impact on the tomato host. However, the section has several shortcomings that could be addressed to strengthen the contextual background and research motivation for readers:

1.       The introduction briefly mentions JA, SA, and ET pathways but could benefit from a more comprehensive description of their roles in plant defense. Including a deeper overview of the function of these phytohormones in plant immunity and why they are critical in the context of RKN infection would enhance the reader's understanding of the study’s focus.

2.       Although the paper aims to explore the role of reactive oxygen species (ROS) and the antioxidant system in the nematode suppression of plant immunity, the introduction does not adequately describe these mechanisms. A brief overview of ROS signaling in plant stress responses and its dual role in immunity would help contextualize the importance of studying this aspect of RKN interaction.

The statistical methods used in this manuscript have several shortcomings that impact the robustness and clarity of the results:

The manuscript uses paired t-tests to compare treated and untreated plants, but this test isn’t ideal for the way data was collected. Paired t-tests are best when measuring the same subject under two conditions, but here, the plants are grown and treated separately. An independent t-test or an analysis of variance (ANOVA) would likely suit the data better, especially for comparisons across different groups and times. Additionally, the manuscript does not discuss whether key assumptions for the t-test, like normality and equal variances, were checked. Skipping these checks could affect the accuracy of the results, especially with the small sample size.

The manuscript shows promise but requires revisions to address these concerns. Enhancing the background information in the introduction and strengthening the statistical methodology will increase the manuscript’s impact and reliability. I recommend a major review.

Author Response

1.The introduction briefly mentions JA, SA, and ET pathways but could benefit from a more comprehensive description of their roles in plant defense. Including a deeper overview of the function of these phytohormones in plant immunity and why they are critical in the context of RKN infection would enhance the reader's understanding of the study’s focus.

2. Although the paper aims to explore the role of reactive oxygen species (ROS) and the antioxidant system in the nematode suppression of plant immunity, the introduction does not adequately describe these mechanisms. A brief overview of ROS signaling in plant stress responses and its dual role in immunity would help contextualize the importance of studying this aspect of RKN interactions

authors: we added the required additions to the Introduction (in yellow in the revised version)

The manuscript uses paired t-tests to compare treated and untreated plants, but this test isn’t ideal for the way data was collected. Paired t-tests are best when measuring the same subject under two conditions, but here, the plants are grown and treated separately. An independent t-test or an analysis of variance (ANOVA) would likely suit the data better, especially for comparisons across different groups and times. Additionally, the manuscript does not discuss whether key assumptions for the t-test, like normality and equal variances, were checked. Skipping these checks could affect the accuracy of the results, especially with the small sample size.

our data measure the same subject (same infection factors, same growth factors) under 2 conditions (plants not treated, plants treated) - what does it mean: "plants are grown and treated separately", we don't understand - untreated and treated plants are grown together under the same conditions - we published many papers using paired t-tests for this kind of comparisons control-treated plants - independent t-test gives the same results for significance of the difference between 2 means (control-treated plants) - we are interested in this significance, NOT if there's difference between different treatments, so we use paired t-tests because replicates come from different experiments which can give quite different ranges of values. With this kind of test, analogous values are confronted.

 Paired t-test assumptions are:

1) Subjects must be independent. Measurements for one subject do not affect measurements for any other subject

2) Each of the paired measurements must be obtained from the same subject

3) The measured differences are normally distributed

Our data are absolutely coherent with these assumptions

Reviewer 2 Report

Comments and Suggestions for Authors

The manuscript investigates how root-knot nematode (Meloidogyne incognita) infections influence the immune responses and antioxidant systems in tomato plants. The study demonstrates that nematode infections suppress the salicylic acid (SA) immune response in tomatoes while leaving the jasmonic acid (JA) pathways largely unaffected.  The study's objectives are clearly defined, and the results are indeed intriguing. However, I have a few suggestions.

In this study, the authors set the sampling times at 3 and 7 days post-inoculation. However, it is unclear why these specific time points were chosen. Typically, studies of this nature include several sampling points to capture a more comprehensive timeline of the infection process. Providing a rationale for selecting only these two time points would enhance the clarity of the experimental design.

To improve clarity, I would recommend creating a hypothesis flowchart. Given that this research is not within my primary area of expertise, I found it challenging to understand the rationale behind the selection of specific genes for RT-PCR analysis. A flowchart could visually illustrate how nematodes suppress SA-mediated defences, stabilise JA pathways, and activate antioxidant responses to reduce oxidative stress, ultimately facilitating successful infection.

Regarding Figure 4, the resolution is insufficient, making it difficult to clearly distinguish the gel bands. Additionally, the quality of the bands appears suboptimal, which may affect the interpretation of the results. Re-running the gels to obtain clearer, well-defined bands is advised. Furthermore, the figure legend currently lacks sufficient detail and should be revised to include a more comprehensive explanation of the results. Enhancing both the resolution and clarity of the figure, along with a more informative legend, would significantly improve the presentation and interpretation of the data.

The study uses t-tests exclusively to compare treated and control groups; however, this may not be adequate given the multiple treatments involved. Without proper adjustments for multiple comparisons, there is a risk of Type I errors. It would be beneficial to employ ANOVA with post-hoc tests for a more robust statistical analysis, ensuring greater accuracy and reliability of the results.

The manuscript should ensure that all scientific names, like Solanum lycopersicum (LINE 398), are italicised consistently. Proper formatting is important for maintaining scientific accuracy. Please review the text to correct any instances where italics are missing.

In terms of English writing, while the manuscript is generally readable and communicates the key findings, a professional proofreading service or a thorough review by a native English speaker with expertise in scientific writing would be advisable to further polish the language and enhance its overall quality.

Comments on the Quality of English Language

In terms of English writing, while the manuscript is generally readable and communicates the key findings, a professional proofreading service or a thorough review by a native English speaker with expertise in scientific writing would be advisable to further polish the language and enhance its overall quality.

Author Response

In this study, the authors set the sampling times at 3 and 7 days post-inoculation. However, it is unclear why these specific time points were chosen. Typically, studies of this nature include several sampling points to capture a more comprehensive timeline of the infection process. Providing a rationale for selecting only these two time points would enhance the clarity of the experimental design.

authors: We added a Figure 5 for providing the required rationale

enzyme activities were detected at 3, 5, and 7 dpi because enzyme activities may change fast; gene transcripts  at 3 and 7 dpi as their changes are slower. Under our experimental conditions, the early stages of infection are considered from 1 to 7 dpi. We wait until the third day after inoculation to allow the brief migratory phase of the sedentary nematodes. Between 3-7 dpi feeding sites are formed and in this period the outcome (compatible/incompatible) of the interaction is decided; plant reactions are the highest in this period. At 5 dpi, a recordable percentage of sedentary forms has been found (Molinari et al. Molecular Plant Pathology 2014, 15, 255-264

Regarding Figure 4, the resolution is insufficient, making it difficult to clearly distinguish the gel bands. Additionally, the quality of the bands appears suboptimal, which may affect the interpretation of the results. Re-running the gels to obtain clearer, well-defined bands is advised. Furthermore, the figure legend currently lacks sufficient detail and should be revised to include a more comprehensive explanation of the results. Enhancing both the resolution and clarity of the figure, along with a more informative legend, would significantly improve the presentation and interpretation of the data.

authors: resolution of the Figure has been increased as suggested - the legend is: " Isoelectrofocusing of root extracts stained for catalase (CAT) and superoxide dismutase (SOD). Root extracts were obtained from non-inoculated (C) and inoculated (N) plants at 5 days post-inoculation with RKNs. Digital images of mini-gels (3.6 cm separation zone) were adjusted to display white enzyme bands on a dark background. Gels were calibrated using a broad pI calibration kit containing proteins with pI values ranging from 3.5 to 9.3". Legends should not contain any explanation of the results but a description of what the reader is looking at. How the gel was obtained is described in Materials and Methods Section. More information is in the Results and Discussion  Sections

The study uses t-tests exclusively to compare treated and control groups; however, this may not be adequate given the multiple treatments involved. Without proper adjustments for multiple comparisons, there is a risk of Type I errors. It would be beneficial to employ ANOVA with post-hoc tests for a more robust statistical analysis, ensuring greater accuracy and reliability of the results.

authors: our data measure the same subject (same infection factors, same growth factors) under 2 conditions (plants not treated, plants treated) - what does it mean: "plants are grown and treated separately", we don't understand - untreated and treated plants are grown together under the same conditions - we published many papers using paired t-tests for this kind of comparisons control-treated plants - independent t-test gives the same results for significance of the difference between 2 means (control-treated plants) - we are interested in this significance, NOT if there's difference between different treatments, so we use paired t-tests because replicates come from different experiments which can give quite different ranges of values. With this kind of test, analogous values are confronted.

 Paired t-test assumptions are:

1) Subjects must be independent. Measurements for one subject do not affect measurements for any other subject

2) Each of the paired measurements must be obtained from the same subject

3) The measured differences are normally distributed

Our data are absolutely coherent with these assumptions

The manuscript should ensure that all scientific names, like Solanum lycopersicum (LINE 398), are italicised consistently. Proper formatting is important for maintaining scientific accuracy. Please review the text to correct any instances where italics are missing.

authors: it has been done

In terms of English writing, while the manuscript is generally readable and communicates the key findings, a professional proofreading service or a thorough review by a native English speaker with expertise in scientific writing would be advisable to further polish the language and enhance its overall quality.

authors: English has extensively been improved (see yellow parts of revised version)

Round 2

Reviewer 1 Report

Comments and Suggestions for Authors

The authors have addressed all my questions; therefore, I recommend the article for publication.

Reviewer 2 Report

Comments and Suggestions for Authors

Based on the authors' response regarding the use of statistical analysis, I may have misunderstood the initial approach. Since the study primarily focuses on testing the expression of specific genes related to nematode infection, I previously thought that a more interconnected statistical approach would be necessary. However, I now understand that the authors' intent was solely to compare differences between control and treatment groups. Given this specific aim, their choice to use paired t-tests is justified, as it effectively highlights the differences between these two conditions.

Apart from the statistical analysis aspect, I have no further concerns. I appreciate the clarification provided by the authors, which has helped resolve my initial queries. I am satisfied with the revisions made and have no additional comments.